# Vast Parameter Space Exploration of the Virtual Brain: A Modular Framework for Accelerating the Multi-Scale Simulation of Human Brain Dynamics

**Michiel van der Vlag** [1,*] , **Lionel Kusch** [2] , **Alain Destexhe** [3] , **Viktor Jirsa** [2] , **Sandra Diaz-Pier** [1] and **Jennifer S. Goldman** [3]

1 SDL Neuroscience, Forschungszentrum Jülich GmbH, Institute for Advanced Simulation, Jülich Supercomputing Centre (JSC), 52428 Jülich, Germany; s.diaz@fz-juelich.de
2 Institut de Neurosciences des Systèmes, Aix-Marseille University, INSERM, 13007 Marseille, France; lionel.kusch@grenoble-inp.org (L.K.); viktor.jirsa@univ-amu.fr (V.J.)
3 Institute of Neuroscience, CNRS, Paris Saclay University, 91190 Gif-sur-Yvette, France; alain.destexhe@cnrs.fr (A.D.); jennifer.goldman@mail.mcgill.ca (J.S.G.)
* Correspondence: m.van.der.vlag@fz-juelich.de

**Abstract:** Global neural dynamics emerge from multi-scale brain structures, with nodes dynamically communicating to form transient ensembles that may represent neural information. Neural activity can be measured empirically at scales spanning proteins and subcellular domains to neuronal assemblies or whole-brain networks connected through tracts, but it has remained challenging to bridge knowledge between empirically tractable scales. Multi-scale models of brain function have begun to directly link the emergence of global brain dynamics in conscious and unconscious brain states with microscopic changes at the level of cells. In particular, adaptive exponential integrate-and-fire (AdEx) mean-field models representing statistical properties of local populations of neurons have been connected following human tractography data to represent multi-scale neural phenomena in simulations using The Virtual Brain (TVB). While mean-field models can be run on personal computers for short simulations, or in parallel on high-performance computing (HPC) architectures for longer simulations and parameter scans, the computational burden remains red heavy and vast areas of the parameter space remain unexplored. In this work, we report that our HPC framework, a modular set of methods used here to implement the TVB-AdEx model for the graphics processing unit (GPU) and analyze emergent dynamics, notably accelerates simulations and substantially reduces computational resource requirements. The framework preserves the stability and robustness of the TVB-AdEx model, thus facilitating a finer-resolution exploration of vast parameter spaces as well as longer simulations that were previously near impossible to perform. Comparing our GPU implementations of the TVB-AdEx framework with previous implementations using central processing units (CPUs), we first show correspondence of the resulting simulated time-series data from GPU and CPU instantiations. Next, the similarity of parameter combinations, giving rise to patterns of functional connectivity, between brain regions is demonstrated. By varying global coupling together with spike-frequency adaptation, we next replicate previous results indicating interdependence of these parameters in inducing transitions between dynamics associated with conscious and unconscious brain states. Upon further exploring parameter space, we report a nonlinear interplay between the spike-frequency adaptation and subthreshold adaptation, as well as previously unappreciated interactions between the global coupling, adaptation, and propagation velocity of action potentials along the human connectome. Given that simulation and analysis toolkits are made public as open-source packages, this framework serves as a template onto which other models can be easily scripted. Further, personalized data-sets can be used for for the creation of red virtual brain twins toward facilitating more precise approaches to the study of epilepsy, sleep, anesthesia, and disorders of consciousness. These results thus represent potentially impactful, publicly available methods for simulating and analyzing human brain states.

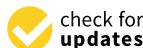



**Keywords:** brain simulation; multi-scale; consciousness; high-performance computing; mean-field model; connectome; brain dynamics; phase transition; brain state

## 1. Introduction

The brain is a multi-scale organ, with relevant scales spanning molecules to synapses, to neurons, to local networks consisting of populations of neurons connected by inter-region tracts, and together forming global brain structures that observe and constrain global brain dynamics. Computational models that connect these biologically relevant scales are an active area of research [1–5]. However, a scale-integrated understanding—relating the actions and nonlinear interactions of microscopic variables to macroscopic observables underlying global brain function—remains empirically challenging to derive from multi-modal experimental measurements [6–8]. Thus, computational models can support progress in bridging our understanding across spatio-temporal scales, with the goal of ameliorating predictions for future experimental studies and, potentially, for personalized medical interventions. Recent progress has been made on the modeling of global phase transitions between unconscious-like (synchronous, regular) and conscious-like (asynchronous, irregular) dynamics that emerge from the same human connectomes, depending on the parameters describing cellular phenomena at the level of spikes [9–12]. Specifically, human, mouse, and primate TVB-AdEx (an implementation of the adaptive exponential (AdEx) model for The Virtual Brain (TVB)) models can describe the emergence of conscious-like asynchronous, irregular activity at global brain scales based on diminished spike-frequency adaptation in simulated waking states due to enhanced acetylcholine concentrations (common to conscious brain states). In contrast, unconscious-like synchronous, regular global brain activity can be simulated by enhancing spike-frequency adaptation, biologically related to diminished neuromodulation during sleep [9,10,13]. Further, cellular hyperpolarization, a mechanism onto which multiple anesthetics converge, can lead to unconscious-like dynamics [14]. However, biophysical-based multi-scale models are complex, requiring substantial computational resources to represent large parameter spaces that describe relevant observables. In addition to studying the behavior of simulations resulting from the modulation of any one variable, it is important to understand the ensemble of parameters and observables, as multiple sets may act redundantly and/or interact nonlinearly. While parameters can be estimated through inference [11,15–18], such methods can be complemented by a better understanding of general parameter spaces of these complex, multi-scale, biophysical human brain models, in order to assess whether previously unconsidered predictions naturally emerge. We have selected the TVB-AdEx model for this benchmarking study due to its significant computational requirements, potential substantial further scientific contribution to theoretical studies of consciousness, and under-study of its behavior arising from different parametrizations. Nevertheless, it is important to note that the presented framework is not restricted to the AdEx model; other neural mass models can be employed in accelerated/ameliorated form, thanks to our modular design.

The present work describes the GPU implementation of the TVB-AdEx, a multi-scale model that describes the population statistics of excitatory and inhibitory AdEx neurons using mean-field approximations with spatial coordinates mapping to human brain regions. The AdEx mean-field model was initially developed to describe the spontaneous and evoked dynamics of local populations of neurons with transfer functions that are semi-analytically derived [19,20]. To form the (human) TVB-AdEx, mesoscopic AdEx mean-fields are placed on and connected by empirical anatomical and diffusion magnetic resonance imaging (MRI) data. The CPU-based (https://github.com/davidaquilue/TVBAdEx_ParSweep/tree/main, accessed on 1 November 2023) version of the TVB-AdEx has previously been used to study scale-integrated mechanisms underlying the emergence of conscious- and unconscious-like brain state dynamics using a local PC, EBRAINS (https://www.ebrains.eu/, accessed on 1 November 2023), an online aggregate of mod-

els [9], and an MPI multi-node CPU setup to study limited compound parametrizations for five parameters known to be fundamental to the behaviour of this AdEx model [10].

Here, we report that our GPU implementation of the TVB-AdEx model substantially ameliorates the original CPU instantiation by accelerating model performance and facilitating post data-analysis. The TVB-HPC framework is not another platform; it represents an expanded framework built upon RateML, TVB's [21] model generator [22]. RateML is performant, modular, reusable, and outperforms solutions such as neurolib [11], FastTVB [23], and Pyrates [24] in terms of magnitude of explorable parameter space and concurrent TVB instances. Specifically, our results show that it is an excellent compute accelerator for TVB-AdEx simulations, allowing for high-dimensional explorations of the parameter space and the identification of parameters acting and interacting to bias brain state dynamics. This work highlights the importance of performant processing for complex brain models, such as biophysical multi-scale TVB-AdEx models, since the dimensionality of parameter space grows easily, quickly turning the problem intractable on previously existing simulation platforms. By applying a wide range of accessible analysis methods, users are further enabled to explore observables describing simulated brain dynamics, including power spectral densities, exhibitory and inhibitory firing rates, the quantification of up and down states, and structure–function relationships, using a repository where these tools are publicly available in EBRAINS (https://wiki.ebrains.eu/bin/view/Collabs/rateml-tvb/Lab, accessed on 1 November 2023). The TVB-HPC project, including the driver and GPU AdEx model, analysis toolkit, and graph script are also available at the following link: https://github.com/DeLaVlag/vast-TVB-space(accessed on 1 November 2023).

We report that a single GPU, an NVIDIA A100 Tensor Core GPU with 40 GB and Compute Unified Device Architecture (CUDA) version 12.2.0, can be employed to explore roughly 17,000 parameter combinations (unique TVB simulations), simulating 68 brain regions for 50,000 simulation steps of 0.1 ms each. Utilizing a configuration comprising 384 nodes—the maximum available nodes for regular production—with four GPUs each facilitated the concurrent execution of approximately 25 million TVB instances. Theoretically, employing all 936 of the GPU-enhanced nodes within the Juwels Booster supercomputing cluster (https://apps.fz-juelich.de/jsc/hps/juwels/booster-overview.html, accessed on 1 November 2023), each with four GPUs connected via NVLink3 (high-speed communication technology), would allow a simulation to concurrently execute approximately 60 million TVB instances, each configured with a distinct set of parameters. The execution of a simulation representing 5 s of biological time would require roughly 8 min to complete for this entire ensemble of instances. Thus, the implementation reported here represents a powerful, personalizable simulation framework to complement and extend multi-scale empirical brain research that is built into EBRAINS tools intended to be useful for fundamental and translational research, with the capacity to inform medical providers in search of precision tools capable of providing multi-scale explanations and predictions for individual human subjects.

## 2. Materials and Methods

The TVB-AdEx model uses mean-field modeling to integrate properties of excitatory and inhibitory AdEx neurons across scales. The populations are connected via a human connectome. It was observed that the macroscopic dynamics resembling brain activity emerge during simulation, with global conscious- and unconscious-like activity emerging from the tuning of parameters representing microscale phenomena [9,10,14].

### 2.1. Mean-Field Model

The TVB-AdEx model is based on a mean-field method capturing the second-order firing rate of an excitatory and inhibitory population of adaptive exponential integrate-and-fire neurons [25] using a Master Equation formalism [19,26], as well as the first order of the adaptive current of the excitatory population [20]. It results in a system of differential equations describing the time evolution of the mean firing rate ($\nu$) of the two populations

($\mu = e, i$), the variance ($c_{\lambda,\eta}$, where $\lambda = \eta$) and covariance ($c_{\lambda,\eta}$, where $\lambda \neq \eta$) of excitatory and inhibitory firing rates ($\lambda = e, i$ and $\eta = e, i$), and the mean adaptive current ($W_\mu$) for the excitatory population.

$$T\frac{d\nu_\mu}{dt} = (F_\mu - \nu_\mu) + \frac{1}{2}c_{\lambda\eta}\frac{\partial^2 F_\mu}{\partial\nu_\lambda\partial\nu_\eta} \tag{1}$$

$$T\frac{dc_{\lambda\eta}}{dt} = \delta_{\lambda\eta}\frac{F_\lambda(1/T - F_\eta)}{N_\lambda} + (F_\lambda - \nu_\lambda)(F_\eta - \nu_\eta)$$

$$+ \frac{\partial F_\lambda}{\partial\nu_\mu}c_{\eta\mu} + \frac{\partial F_\eta}{\partial\nu_\mu}c_{\lambda\mu} - 2c_{\lambda\eta} \tag{2}$$

$$\frac{\partial W}{\partial t} = -W/\tau_w + b_e\nu_e + a_e(\mu_V(\nu_e, \nu_i, W) - E_{L,e}) \tag{3}$$

where $T$ is the time constant of the firing rate equations and covariance equations and $F_\mu$ is the transfer function. $\tau_w$, $b_e$, and $a_e$ are, respectively, the time constant, the spike-frequency, and subthreshold of the adaptation. $E_L$ is the leak reversal potential and $\mu_V$ is an estimation of the average voltage of the excitatory population.

A stochastic equation, denoted as Equation (4), is employed to represent an Ornstein–Uhlenbeck (OU) process, capturing the random fluctuations of the mean firing rate, commonly known as the noise. This process is derived by introducing a mean-reverting term to standard Brownian motion, a stochastic process similar to the random movement of particles in a fluid or gas. This mean-reverting term is considered to be a stabilizing component to the motion, resulting in a tendency to return to a central value. The computation of the derivative of this stochastic equation is stored in GPU memory to be available for the next time-step.

$$\tau_{OU}\frac{dOU_t}{dt} = (\mu - OU_t) + \sigma\, dW_t \tag{4}$$

where $\mu$ (=0.0) is the mean of the noise, $\sigma$ (=1.0) is the variance of the noise, and $dW_t$ is a Wiener process (Brownian motion). $\tau_{OU}$ represents the time constant of the noise. The $dW_t$ is represented by a Gaussian noise implemented with the CUDA Random Number Generation (CURAND) library, making use of a XORWOW generator, a variation on the XOR shifting generators. The next number in the sequence is generated by repeatedly taking the exclusive OR of a number with a bit-shifted version of itself [27].

*2.2. Transfer Function*

The transfer function (TF) gives the instantaneous mean firing rate following the state of the populations and the external inputs [19,20,26].

$$F_{\mu=\{e,i\}} = F_{\mu=\{e,i\}}(\nu_e^{k,tot} + wNoise.OU_t^k + c_{F_{\mu,e}}, \nu_i^k + c_{F_{\mu,i}}, W_\mu^k) \text{ for brain region } k, \tag{5}$$

where $\nu_e^{k,tot} = g\left(\sum_{68}^{j=1} u_{kj}\nu_{e_j}(t - \tau_{kj})\right)$ is the weighted sum of all firing rates of all brain regions, input to the region $k$. The weights ($u_{kj}$) are defined by the connectome. The delays ($\tau_{k,j}$) equal the track lengths between the region $k$ and region $j$ divided by the speed. $OU_t^k$ is the noise associated with the brain region (see above). $c_{F_{\mu,\lambda}}$ is either the constant excitatory or inhibitory input to the population.

One of the principal novel aspects of this formalism is that the mean-field depends on the specification of the transfer function $F_{\mu=\{e,i\}}$, which can be obtained according to a semi-analytical approach [19], and thus can be potentially applicable to many different models [28]. In this approach, the transfer function is numerically fit to single-cell responses using an analytic template depending on three parameters: the mean voltage ($\mu_V$), its standard deviation ($\sigma V$), and its time correlation decay time ($\tau_V$). Additionally, a voltage threshold ($V_{thre}^{eff}$) is estimated from the properties of an individual neuron taken as a second-

order polynomial (depicted in Table 1) of the previous quantities ($\mu_V, \sigma V, \tau_V$). Using the assumption that the voltage membrane of the population follows a normal law and the phenomenological voltage threshold, it is possible to estimate the mean firing rate of the population, i.e., the output of the transfer function [19,20].

**Table 1.** Polynomial of the phenomenological voltage threshold for the transfer function (mV).

| Cell Type | $P_0$ | $P_{\mu V}$ | $P_{\sigma V}$ | $P_{\tau NV}$ | $P_{\mu 2}$ | $P_{\sigma 2}$ | $P_{\tau NV2}$ | $P_{V\sigma}$ | $P_{\mu V_\tau N}$ | $P_{\sigma V\tau N}$ |
|---|---|---|---|---|---|---|---|---|---|---|
| RS-Cell | −48.9 | 5.1 | −25.0 | 1.4 | −0.41 | 10.5 | −36.0 | 7.4 | 1.2 | 40.7 |
| FS-Cell | −51.4 | 0.4 | −8.3 | 0.2 | −0.5 | 1.4 | −14.6 | 4.5 | 2.8 | 15.3 |

To advance beyond previously explored corners of the phase space for the multi-scale TVB-AdEx model with the published implementation [9,10], prohibitively large computational resources would have been required. Therefore, we have implemented the TVB-AdEx model for GPUs, which can be more efficient for computations that are embarrassingly parallel. Parameterization in general is embarrassingly parallel due to an absence of dependency between simulations; each parametric combination does not depend on other concurrent simulations. A GPU is therefore an excellent choice for acceleration; the data latency of the individual simulations is largely covered by the many concurrent simulations. The many smaller computational units of the GPU, in comparison to a CPU, have also proven to be sufficient for TVB simulations.

### 2.3. TVB-AdEx to GPUS

To accommodate the intricate regimes associated with various parameter settings within the TVB-AdEx model, a GPU-accelerated version has been developed. The TVB model generator, RateML, offers the capability to create customized rate-based or mean-field models. In the standard workflow of this tool, an XML file can be populated with TVB generic properties, including the derivative functions of the model. This file is then translated into a fully fledged TVB model or a GPU model and driver for the execution of the model. Because the AdEx model has a complex transfer function, in describing the firing rate of the neuron populations, the regular usage of RateML is not possible. However, the GPU model introduced here is based on the blueprint for TVB GPU models, manually annotated with the added model complexity, including the function calls to the transfer function (*zerlaut.c*). The transfer function itself is specified in a separate header file (*zerlaut.h*). Next to a GPU model which supports parameter sweeps, the model driver (*model_driver_zerlaut.py*) object is also generated when invoking RateML. Furthermore, this driver has been manually annotated for the computation and comparison of the functional connectivity (FC) to an externally obtained FC.

The model driver and model for the TVB simulation using the GPU AdEx can be initiated on any CUDA-capable GPU device. An example of the execution command is as follows:

```
python model_driver_zerlaut_mpi.py -s0 6 -s1 6 -s2 6 -s3 6 -s4 6
-n 5000 -dt 0.1 -vw -cs 3 -cf connectivity_zerlaut_68.zip
-ff pearson_0.4_72_-64_-64_19 -gs
```

A number of simulator settings are exposed to the user on command line. Many more settings are exposed to the user but the most important are highlighted. The *s0–s4* are the flags that set the resolution for the sweep parameters of the specified parameters. That is, *-s0* 6 generates six values equally spaced within the defined range for the first parameter of the space to be explored. Normally, the lower and upper bounds of the parameters to sweep can be indicated in the XML file or can be changed in the driver file. This will result in the generation of an array of all the possible parameter combinations, representing the total work-items. The work-items correspond to the grid size indicating the total threads which

spawn on the GPU. Each thread spawned is a TVB simulation with a unique parameter combination. The two-dimensional grid for the GPU threads is determined recursively according to the number of work-items. The program will try to fit the work-items in blocks of $32 \times 32$ threads and increase the number of blocks, alternating for the x- and y-axis dimensions in order to find the optimal population of the GPU.

The simulation of the GPU consists of two loops, an inner-loop, which determines the number of iterations of the simulation on the GPU, and an outer-loop, which determines the number of times a GPU-instance spawns. The $-n$ sets the number of simulation steps for the outer-loop and determines the number of times a GPU-instance is spawned. The steps for the inner-loop are computed by dividing the period of the time-series by the delta-time, $-dt$. This influences the precision of the computation of the time derivatives of the dynamical system, as the end result is the average of the number of these steps, and reduces the number of times the GPU has to off-load its memory contents to the disk. Alongside the spawning of multiple TVB simulations on the GPU itself, the driver makes use of 32 streams for the iterations of the outer-loop, executing the GPU-instances concurrently as well.

*2.4. Output*

After each iteration of the outer-loop, the set of observables is written to memory. The number of observables can be manually increased by uncommenting the lines added for the default state variables or by adding your own observable by setting [...] to a valid expression in the driver file:

```
//  tavg(i_node + 0 * n_node) += C_ee/n_step;
    tavg(i_node + 1 * n_node) += [...]/n_step;
```

In this scenario, the derived observable is the average across all the inner steps, for one iteration of the outer-loop. It is noteworthy that the characterization is not constrained exclusively to the average value.

The connectivity file can be specified with the $-cf$ flag. If not specified, a connectome from the standard TVB library can be selected instead. In the example, a distinctive connectome, *connectivity_zerlaut_68.zip*, originally employed in the scale-integration studies [9] is utilized. The values for the weights of the connectome, used to determine the temporal connections of the regions, are averaged before simulation. The $-cs3$ flag specifies the conductance speed of the connectome in $m/s$ and also determines the depth of the temporal buffer.

The GPU obtains the functional connectivity (FC) by computing the covariance of the resulting time-series for all concurrent TVB simulations, which, in this study, is used for structure-to-function analysis (FCSC). The FCSC, the correlation with the structure connectivity (SC), represented by the properties of the connectome and the obtained covariance matrix, is determined with the Pearson correlation. The driver can also perform a comparison to an externally obtained reference FC matrix by correlating it to the computed FC of the simulation, also by making use of the Pearson coefficient. The file for input can be specified with the $-ff$ flag. The framework also integrates the fMRI_Obs library from https://github.com/dagush/fMRI_Observables (accessed on 1 November 2023), enabling run-time computation of the phase functional connectivity dynamics (phFCD) for comparison. Optionally, a Balloon–Windkessel GPU kernel [29] can be applied to obtain the BOLD time-series, enabling the comparison with fMRI imaging as well.

*2.5. MPI*

The framework is enhanced with a message passing interface (MPI) (https://www.mpi-forum.org/docs/, accessed on 1 November 2023), enabling the simultaneous execution of multiple parallel processes distributed across compute nodes within a high-performance computing cluster. This utilization of MPI, a standard communication protocol in parallel computing, facilitates enhanced computational efficiency by harnessing the collective

processing power of multiple nodes. A single GPU can be assigned to each MPI process and, as there are no dependencies between the execution of the simulations with different parameters, all processes can utilize these resources in parallel. For example, the command *srun -N2 –ntasks-per-node=4* instantiates multiple MPI processes across 2 nodes, each executing many TVB simulations concurrently on independent GPUs. In this example, the MPI world consists of 8 processes, each assigned a single GPU. The computational workload, referred to as work-items, will be evenly distributed among all available GPUs. This distribution is achieved by reshaping the array in accordance with the world size and subsequently allocating each rank a corresponding portion of the workload. When multiple GPUs are specified for execution, the driver will sort and gather the results from each instance in the MPI world and output the ten best-fitting simulations. The sorting is enabled using the *-gs* flag, which is short for grid search. The *-v* flag sets the output to verbose and *-w* saves the 10 best time-series and their parameters and fitness of each world to a file. When run in verbose mode, the driver outputs information about the simulation and GPU memory allocation for rank 0 only. It also will also output detailed error information on memory allocation or run-time errors, i.e., too large parameter sets or grid allocation failures.

The simulation uses a standard forward Euler method to approximate the time derivative integral, and the linear coupling is applied to globally connect the brain regions for computing the brain dynamics in time, as implemented by the standard TVB library.

### 2.6. Analysis Metrics

The framework adapts and incorporates an extensive array of tools [9,10] (as listed in Table 2 below) in order to analyze the results from the GPU simulations. The outcomes derived from the analysis are systematically stored in a database using the Python library "sqlite3"(https://www.sqlite.org/index.html, accessed on 1 November 2023), where they are formatted and organized using Structured Query Language (SQL).

**Table 2.** The incorporated analysis metrics showing the function name and description.

| Metric | Description |
| --- | --- |
| mean_FC | Average FC from Pearson corr. of time-series firing rate |
| mean_PLI | Average PLI between brain regions. |
| mean_UD_duration | Mean duration of UP and DOWN states |
| psd_fmax_ampmax | PSD frequency peaks and amplitude |
| fit_psd_slope | Fits $log(PSD) = log(\beta/f^{\alpha})$ |

The phase locking index (PLI) is a statistical measure frequently employed in the fields of neuroscience and signal processing. Its primary purpose is to evaluate the level of phase synchronization or phase consistency observed between the different nodes of the time-series, notably in the context of electroencephalography (EEG) and magnetoencephalography (MEG) data analysis [9,30,31]. The PLI serves as a quantification tool for assessing the degree to which the phase angles of two signals exhibit synchronization tendencies over a period of time.

The terms "up" and "down" states are utilized to denote distinct patterns of neuronal activity within the brain. These patterns are of particular significance in the realms of neural oscillations and sleep research. They are closely linked to the sleep–wake cycle and serve as vital components for comprehending the dynamics of brain function throughout periods of sleep and wakefulness.

The acronym "PSD" refers to "Power Spectral Density" in the field of signal processing. This fundamental concept serves as a means to articulate the manner in which power or energy is distributed within a signal concerning its frequency components. PSD plays a

pivotal role by furnishing essential insights into the spectral characteristics and frequency content of the signal under examination.

The PSD is approximated by the following function:

$$log(PSD) = log(\beta/f^{\alpha})$$

in which $f$ is the frequency; $\alpha$ and $\beta$ are the parameters that affect the shape and the amplitude of the function, respectively. The goal of fitting the PSD to this particular function is to find the values of the parameters $\alpha$ and $\beta$ that best describe the observed data. By applying a logarithm to both sides of the equation, a linear relationship is obtained, which is easier to work with mathematically. By fitting the PSD data to this function and determining the values of $\alpha$ and $\beta$, we can gain a better understanding of how the power in the signal is distributed across different frequencies and the characteristics of the spectral density.

## 3. Results

To reflect multi-scale neural activity, biophysical models of the brain can become complex, with dynamical behavior dependent on each parameter as well as potentially nonlinear interactions between parameters. Toward progress in understanding the integration of neural phenomena across scales, we have constructed a multi-GPU implementation of the TVB-AdEx model: a multi-scale model summarizing the statistics of excitatory and inhibitory populations of spiking neurons [19,20] that has demonstrated utility in modeling transitions between conscious- and unconscious-like brain dynamics [9,14,19,20]. Specifically, by coupling mean-fields representing brain regions by human connectome data [9], it was observed that macroscopic dynamics resembling brain activity emerge during simulation, with globally conscious- and unconscious-like activity dependent on parameters representing microscale phenomena known to be related to sleep–wake cycles and the actions of anesthetic agents [9,10,14].

To further advance beyond previously explored corners of the phase space for the multi-scale TVB-AdEx model with the published implementation [9,10,14], prohibitively large computational resources would have been required. Therefore, we have implemented the TVB-AdEx model for GPUs, which can be more efficient for computations that are embarrassingly parallel. Parameterization in general is embarrassingly parallel due to an absence of dependency between simulations; each parametric combination does not depend on other concurrent simulations. The GPU can therefore be an excellent choice for acceleration; the data latency of the individual simulations is largely covered by the many concurrent simulations.

For a detailed description of the implementation of our TVB-HPC framework, see Materials and Methods. Briefly, RateML's tools were modified and manually annotated to produce the simulation code, its dependencies, and outputs, including the transfer function and analysis pipelines. The framework can be initiated on any CUDA-capable GPU device.

To determine whether the GPU model robustly reproduces the behavior of the previously published CPU implementation [9,10,14], simulations involving multiple parameter sets are conducted on the platform. These simulations are executed without introducing noise, aiming to assess the framework's ability to accurately replicate CPU results. The evaluation is based on both the mean-squared error and the correlation coefficient of time-series generated with the same initial conditions and kernels, which, for all parameter combinations, are observed to have differences that do not exceed $1 \times 10^{-9}$ and 1.0, respectively. This implies that the CPU and GPU models generate identical results.

### 3.1. Functional Connectivity

To showcase the efficacy of the functional connectivity (FC) comparison, a series of simulations are carried out. The CPU data are analogous, i.e., an externally acquired EEG scan representing four distinct types of brain states. The goal is to hypothetically create a "virtual brain twin" of the CPU data by obtaining a GPU parametrization that aligns

with the observed CPU data. The externally obtained FCs are compared to the online computed FC from the simulation of the TVB-AdEx model on the GPU in order to find the best matching sets of parameters reproducing the model's behaviour. Parameter sets with values $S_0(g = 0.0, b_e = 0.0), S_1(g = 0.3, b_e = 24), S_2(g = 0.4, b_e = 72), S_3(g = 0.4, b_e = 120)$ are considered. For a full description on how to make use of this comparison, see Section 2. The reported fitness represents the Pearson correlation between the external (CPU) and the FC computed online with a GPU, i.e., Equation (6):

$$\rho_{xy} = \frac{\text{Cov}(x, y)}{\sigma_x \sigma_y}. \tag{6}$$

The results of the four FC comparisons conducted on data from simulations of the model for different states are presented in Figure 1. In total, 16,384 parameter combinations are considered. The results provide a visual representation of the distribution of the top 10 distinct solutions across the parameter space for the different parameter sets. In order to enhance the visualization, a solution sphere has been constructed, with the radius of each dimension represented by the respective range values. The results indicate an overall parameter match, achieving approximately 70% similarity with the reference FC.

It is essential to note that FC comparisons lack a unique solution, due to the stochastic nature of the AdEx model as Brownian noise mimics neural fluctuations and activates the model. The GPU, in this context, showcases its ability to discover multiple characterizations for the model that closely align with the FC. In studies involving FC comparisons, a intersession correlations are deemed high when exceeding 70% [32]. The obtained results affirm that the GPU can successfully replicate such outcomes, producing highly correlated FC results for externally acquired data-sets.

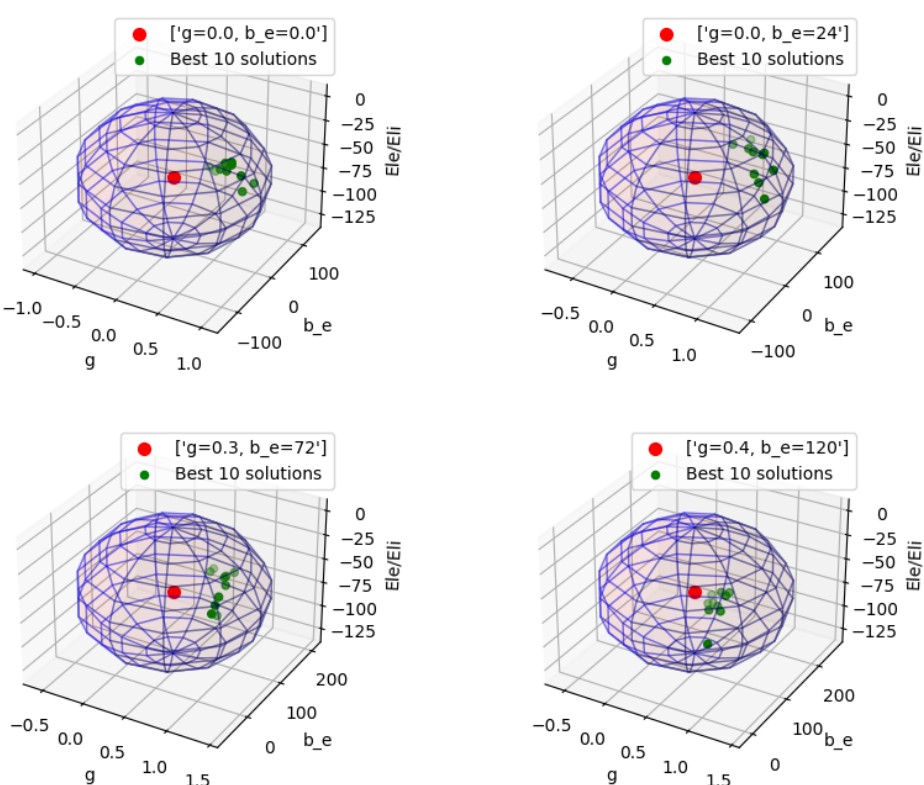

**Figure 1.** Visual representation of the best 10 solutions; fitting GPU TVB-AdEx to CPU TVB-AdEx reference FC matrices for four different parameter sets indicates good mapping between stochastic models for analyses of brain dynamics with no unique solution. The value of $El_e$ and $El_i$ is $-64$ mV for the reference FC. The 3D spheres depict the solution space of the GPU TVB-AdEx and the center red dot represents the input parameters of the reference CPU TVB-AdEx FC.

### 3.2. Vast Parameter Space Exploration and Analysis

Leveraging the computational resources of the Juwels Booster compute cluster at the Forschungszentrum Jülich empowers extensive parameter space exploration, enabling a comprehensive investigation across a wider spectrum of parameters, as described in Table 3. This extensive computational capability has allowed us to not only validate the model, but to also investigate the extensive properties of the associated parameter space. These considerable hardware resources have also facilitated the search for transitions in simulated brain dynamics, the identification of peaks in the power spectrum, the characterization of up and down states typical for synchronous behavior (for a full description, see Table 2), and the examination of factors such as the influence of the velocity of action's potential propagation through the connectome on adaptation (for a full description, see Table 3).

**Table 3.** Parameters targeted for vast exploration.

| Name | Range | Resolution | Description |
| --- | --- | --- | --- |
| $g$ | $[0.1, 0.9]$ | 8 | Coupling strength connectome |
| $b_e$ | $[0, 100]$ | 8 | Spike-frequency adaptation [pA] |
| $wNoise$ | $[1 \times 10^{-9}, 1 \times 10^{-4}]$ | 4 | Scaling weight of noise |
| $speed$ | $[1, 7]$ | 4 | Connectome speed [m/s] |
| $\tau_w$ | $[250, 750]$ | 4 | Adaptation time constant exc. neurons [ms] |
| $a_e$ | $[-10, 20]$ | 4 | Subthreshold adaptation conductance [nS] |
| $c_{F_e,e}$ | $[0.3 \times 10^{-3}, 0.5 \times 10^{-3}]$ | 2 | External input [Hz] |
| $c_{F_e,i}$ | $[0, 0.5 \times 10^{-3}]$ | 2 | External input [Hz] |
| $c_{F_i,e}$ | $[0.3 \times 10^{-3}, 0.5 \times 10^{-3}]$ | 2 | External input [Hz] |
| $c_{F_i,i}$ | $[0, 0.5 \times 10^{-3}]$ | 2 | External input [Hz] |

#### 3.2.1. The Effect of Modulating Coupling and Spike-Frequency Adaptation

The TVB-AdEx model can be used to study different types of brain dynamics associated with different states of consciousness; asynchronous irregular dynamics associated with wakeful, conscious awareness as well as synchronous, regular dynamics associated with unconscious brains states. To validate whether the GPU model can faithfully represent dynamics associated with these regimes through the modulation of variables previously studied [10], the global coupling, $g$, and spike-frequency adaptation, $b_e$, parameters were swept over the aforementioned ranges (Table 4). In Figure 2, time-traces display the raw output for a sweep of $g$ and $b_e$ in the ranges of $[0.3, 0.9]$ and $[0, 120]$, respectively, for a total of 36 parameter combinations, showing the model's ability to simulate transitions between disordered asynchronous and more ordered synchronous behavior. These results demonstrate the interaction between coupling and spike-frequency adaptation, showing the most synchronous-regular activity in the bottom-right-corner panel, where both parameters are maximal, thus further replicating the previously established behavior of the CPU model using GPUs for the TVB-HPC framework.

Furthermore, Figure 3 shows that the excitatory firing rate, $F_e$, is more sensitive to the spike-frequency adaptation, and the excitatory firing rate variability is higher for lower coupling values. In contrast, the inhibitory firing rate $F_i$ is influenced by both coupling and spike-frequency adaptation. Elevated coupling or the spike-frequency adaptation leads to an increase in both the mean and the variance of $F_i$. These results concur with the time-series displayed in Figure 2, illustrating that elevated coupling values combined with high spike-frequency adaptation (bottom right) correspond to heightened variance and mean values of $F_i$. Visa versa, when both the spike-frequency adaptation and the coupling are low, the mean values for $F_i$ are lower.

**Table 4.** Memory utilization and corresponding number of unique TVB simulations (#TVBs) for the GPU for strong scaling.

| Nodes | Memory (GB) | #TVBs |
|---|---|---|
| 1 | 36,330 | 16,384 |
| 2 | 18,622 | 8192 |
| 4 | 9768 | 4096 |
| 8 | 5336 | 2048 |
| 16 | 3120 | 1024 |

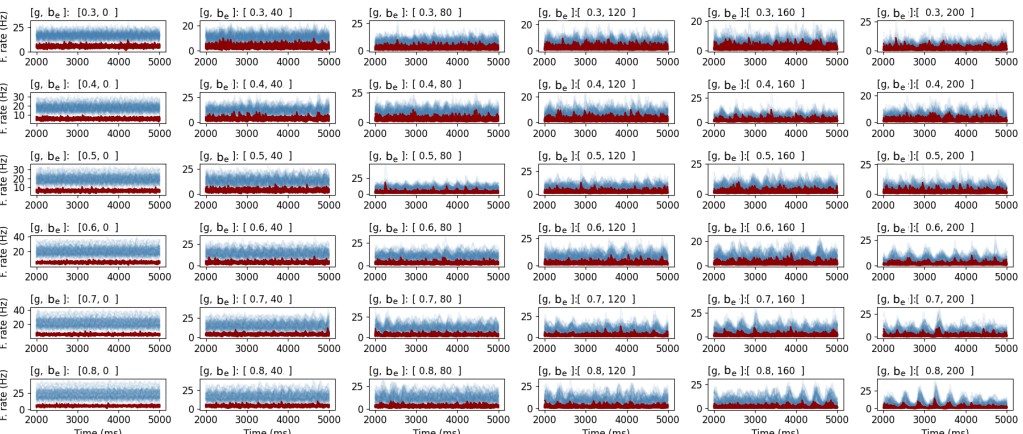

**Figure 2.** Results of a $6 \times 6$ parameter sweep for global coupling, $g$, and spike-frequency adaptation, $b_e$, parameters. The exhibitory firing rate, $F_e$, is depicted in red and the inhibitory firing rate, $F_e$, in blue. The $g$ and $b_e$ parameters are swept for ranges $[0.3, 0.9]$ and $[0, 120]$, respectively. Results show the output time-series for each individual set of parameters printed above each plot. The plots show the interaction of $g$ and $b_e$, resulting in a gradual transition from asynchronous (top-left corner) to synchronous behavior (bottom-right corner) for the GPU TVB-AdEx model.

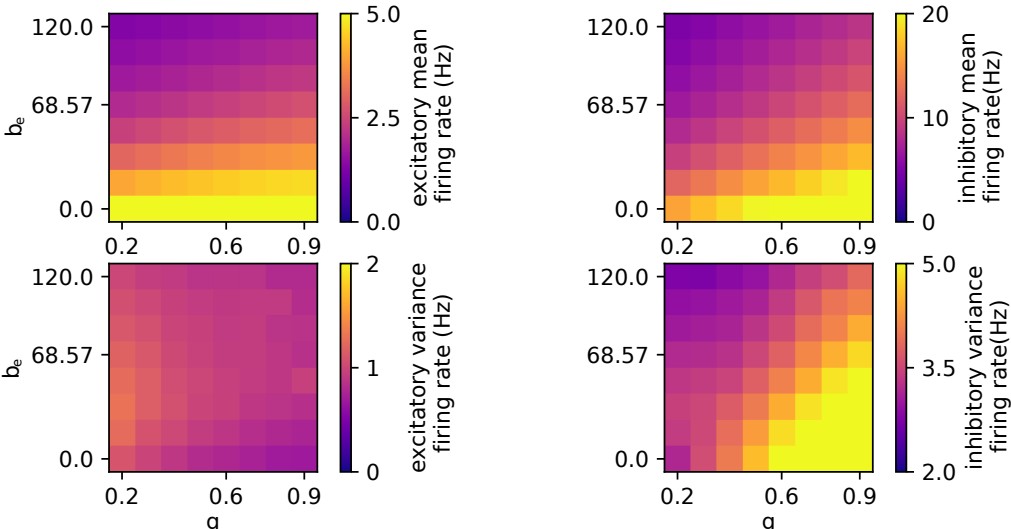

**Figure 3.** Parameter exploration for global coupling and spike-frequency adaptation. The other parameters are as follows: *wNoise*: $1.0 \times 10^{-4}$, *speed*: 3.0, $\tau_w$: 417, $a_e$: 0.0, $c_{F_{e,e}}$: $0.5 \times 10^{-3}$, $c_{F_{e,i}}$: 0.0, $c_{F_{i,e}}$: $0.5 \times 10^{-3}$, $c_{F_{i,i}}$: 0.0.

### 3.2.2. The Effects of Modulating the Adaptation Time Constant

As depicted in Figure 4, an increase in the adaptation time parameter $\tau_w$ yields several effects within the model. Specifically, it leads to a reduction in the mean firing rate, enhances

the linearity of the power–frequency relationship, diminishes the peak frequency value, and promotes an increase in correlation with the structural connectivity (SC).

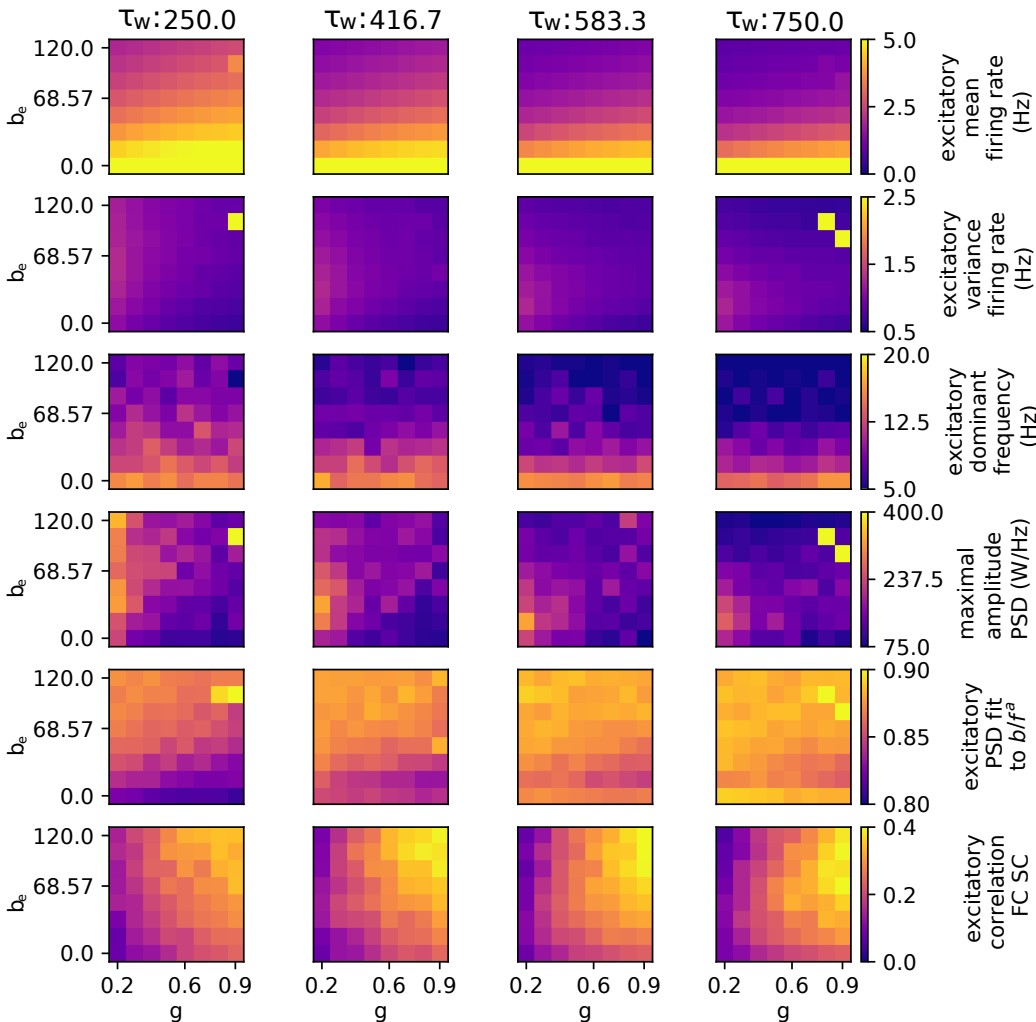

**Figure 4.** Parameter exploration for global coupling and spike-frequency adaptation when modulating the adaptation time constant $\tau_w$. The other parameters are as follows: *wNoise*: $1.0 \times 10^{-4}$, *speed*: 3.0, $a_e$: 0.0, $c_{F_{e,e}}$: $0.5 \times 10^{-3}$, $c_{F_{e,i}}$: 0.0, $c_{F_{i,e}}$: $0.5 \times 10^{-3}$, $c_{F_{i,i}}$: 0.0.

It is observed that, for three specific values ($\tau_w = 250, g = 0.90, b = 102$; $\tau_w = 750.0$, $g = 0.90, b_e = 85$ and $\tau_w = 750.0, g = 0.8, b = 102$), the mean firing rate exceeds 5.0 Hz and the variance exceeds 2.5 Hz. In this case, some brain regions are attracted toward the fixed point around 200 Hz due to the stochastic nature of the model. The outliers depicted in the subsequent figures share a common explanation.

### 3.2.3. The Effects of Modulating Excitatory Subthreshold Adaption Conductance

The occurrence of activity at $a_e = -10$ pA, as depicted in Figure 5, may be attributed to the fact that, during this particular condition, the adaptation acts as an excitatory mechanism due to negative values. When $a_e$ reaches $-10$, an issue arises where excitation becomes excessively pronounced, resulting in all regions exhibiting excitatory activities exceeding 100 Hz, a level deemed excessively high and potentially related to paroxysmal activity and seizure dynamics. However, the model lacks precision when dealing with high-firing-rate activity and lacks an explicit mechanism for transitioning between high- and low-activity states. The parameter $a_e$ introduces a bifurcation effect, specifically impacting the mean results as they fluctuate between higher and lower values at $a_e = -10$. This effect

is distinct from the bifurcation observed with parameters $b_e$ and $g$, which occur at higher values of $b_e$.

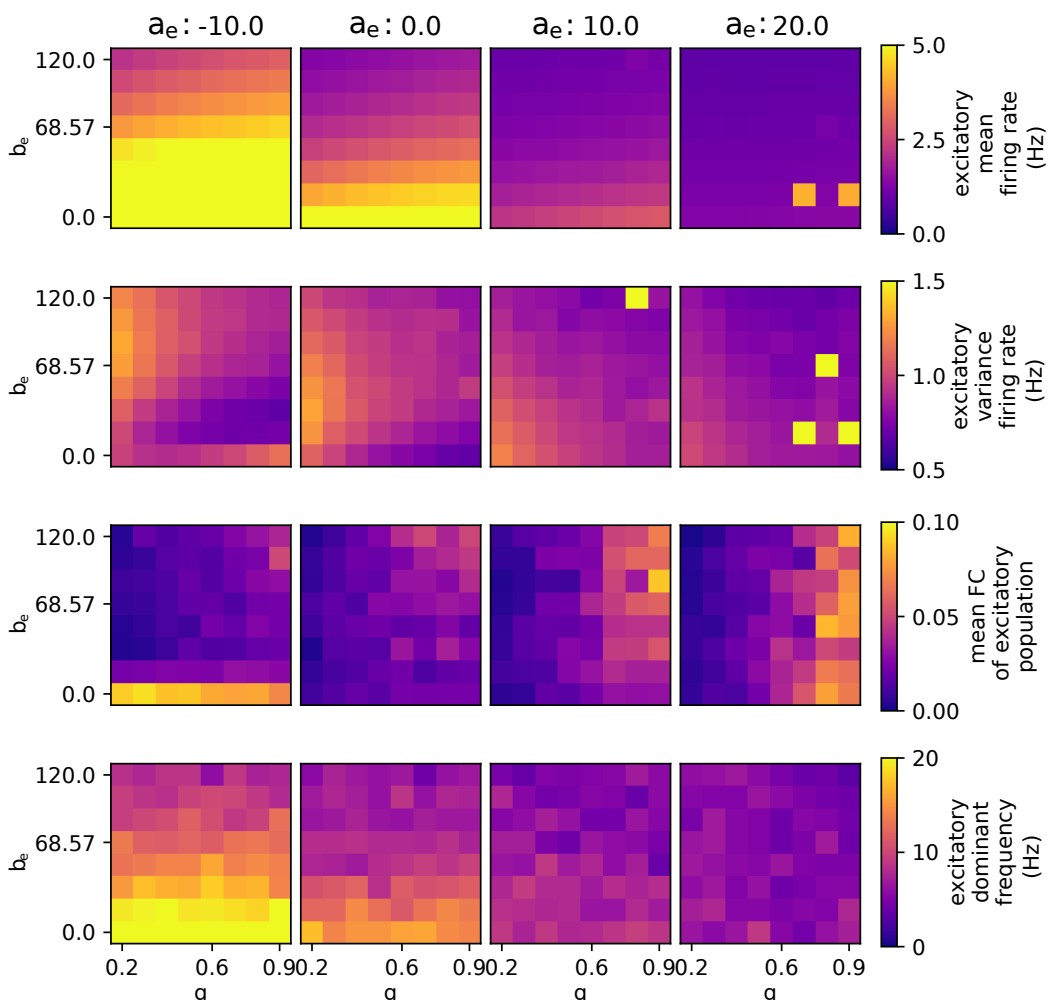

**Figure 5.** Parameter exploration for global coupling and spike-frequency adaptation when modulating the subthreshold adaptation conductance, $a_e$. The other parameters are as follows: *wNoise*: $1.0 \times 10^{-4}$, *speed*: 3.0, $\tau_w$: 417, $c_{F_{e,e}}$: $0.5 \times 10^{-3}$, $c_{F_{e,i}}$: 0.0, $c_{F_{i,e}}$: $0.5 \times 10^{-3}$, $c_{F_{i,i}}$: 0.0.

Furthermore, increasing the value of $a_e$ leads to several outcomes, including a reduction in firing rates, decreased variability, diminished impact of parameter $b_e$, a decrease in the dominance of certain frequency components, and an increase in the correlation between different regions.

### 3.2.4. The Effects of Stimulating External Excitatory and Inhibitory Populations

As depicted in Figure 6, in the context of the model's dynamics, stimulating the excitatory population leads to a shift in mean activity levels, causing transitions both upward and downward, while concurrently reducing the correlation with the structural connectivity (SC). Conversely, stimulating the inhibitory population results in an increase in the mean excitatory firing rate and a decrease in the global spectral frequency. It is important to note that the excitation of both the excitatory and inhibitory neuronal populations collectively influences the maximum peak of variance within the excitatory population.

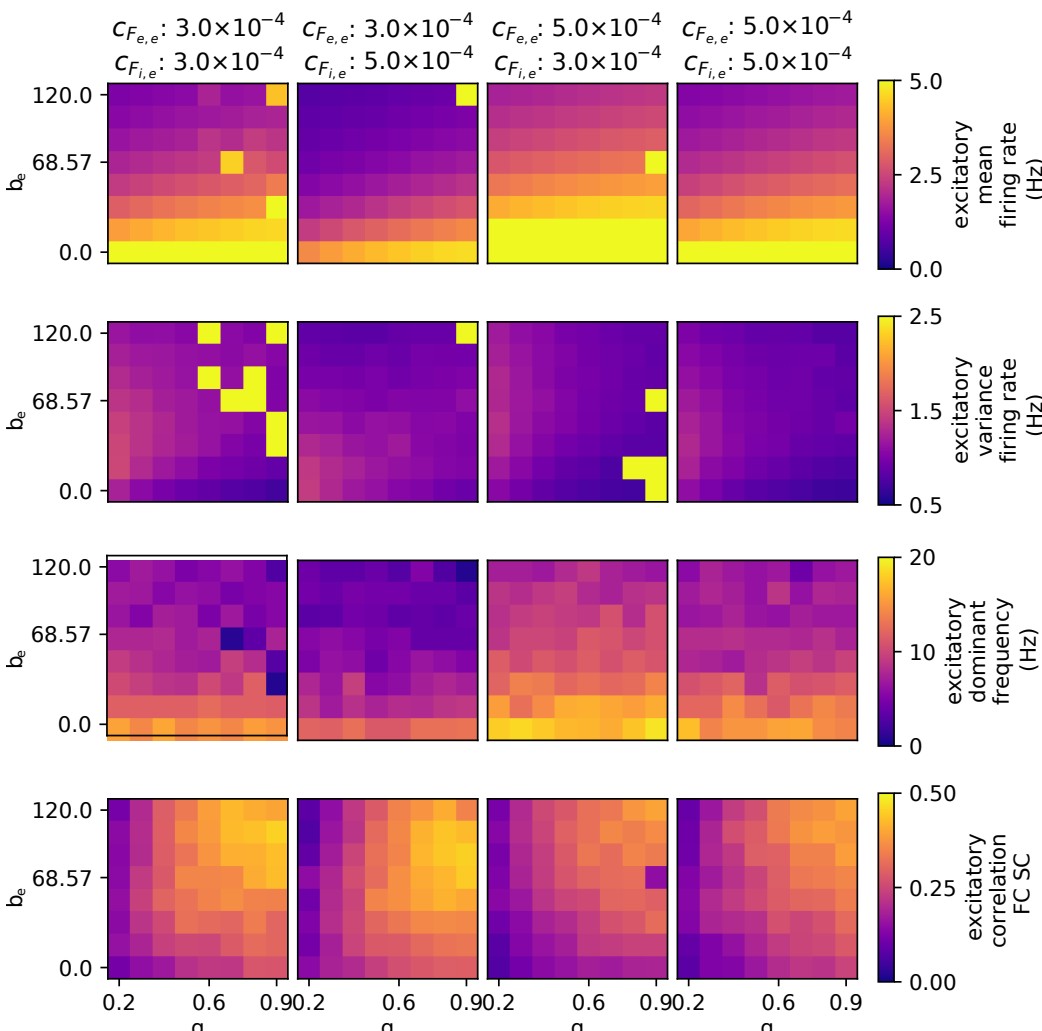

**Figure 6.** Parameter exploration for global coupling and spike-frequency adaptation in the context of different external excitatory inputs. The other parameters are as follows: *wNoise*: $1.0 \times 10^{-4}$, *speed*: 3.0, $\tau_w$: 417, $a_e$: 0.0.

3.2.5. Pathology and the Effect of Modulating the Propagation Speed of Action Potentials through the Connectome

One intriguing aspect of mean-field models obtained through biophysical approaches is their potential utility in identifying abnormal patterns or irregularities associated with pathologies. The TVB-AdEx could potentially be used to investigate seizure activity related to epilepsy, for instance, due to the hyperactive or hypersynchronized states of the model [9] as the dynamical landscape of the model comprises a pathological fixed point around 190 Hz [10]. This is a point where neurons fire directly after their refractory period. The vast parameter sweep can be used to chart the landscape and find the conditions under which the mean-fields of the network settle at this pathological fixed point. The occurrence of reaching this fixed point can be localized by examining the conditions for a jump in the dynamics to high-frequency activity. Disconnected models and models with a lower global coupling have a higher probability of reaching this fixed point [10]. It is to be expected that the fixed point is less likely to be reached if the coupling is stronger.

When examining the effects of modulating the speed of action potential propagation through the connectome, it becomes apparent that when the speed attains a value of 1.0 m/s, particular regions display varying probabilities to visit the paroxysmal fixed point and display a high-firing-rate activity exceeding 100 Hz. This observation is visually depicted in Figure 7. When quantifying the instances in which a high firing rate is observed

exclusively in single regions, we observe the following: The left parahippocampal region visits the paroxysmal fixed point for all 16,384 parameter combinations when the speed attains a value of 1.0 m/s. Furthermore, the subsequent four maximum counts are recorded: 6500 occurrences for the left entorhinal region, 3201 occurrences for the left inferiorparietal region, 1866 occurrences for the right parahippocampal region, and 830 occurrences for the right entorhinal region. Intriguingly, the left parahippocampal region stands out with the highest input weights and has the strongest connection with the left entorhinal region. This observation may suggest that highly connected regions of the model are implicated in the generation of simulated epileptic seizures. This specific observation is only found when the speed property of the connectome, which globally influences the connections between regions, is low and affirms the assumption that less connected brains have a higher probability of reaching the pathological fixed point.

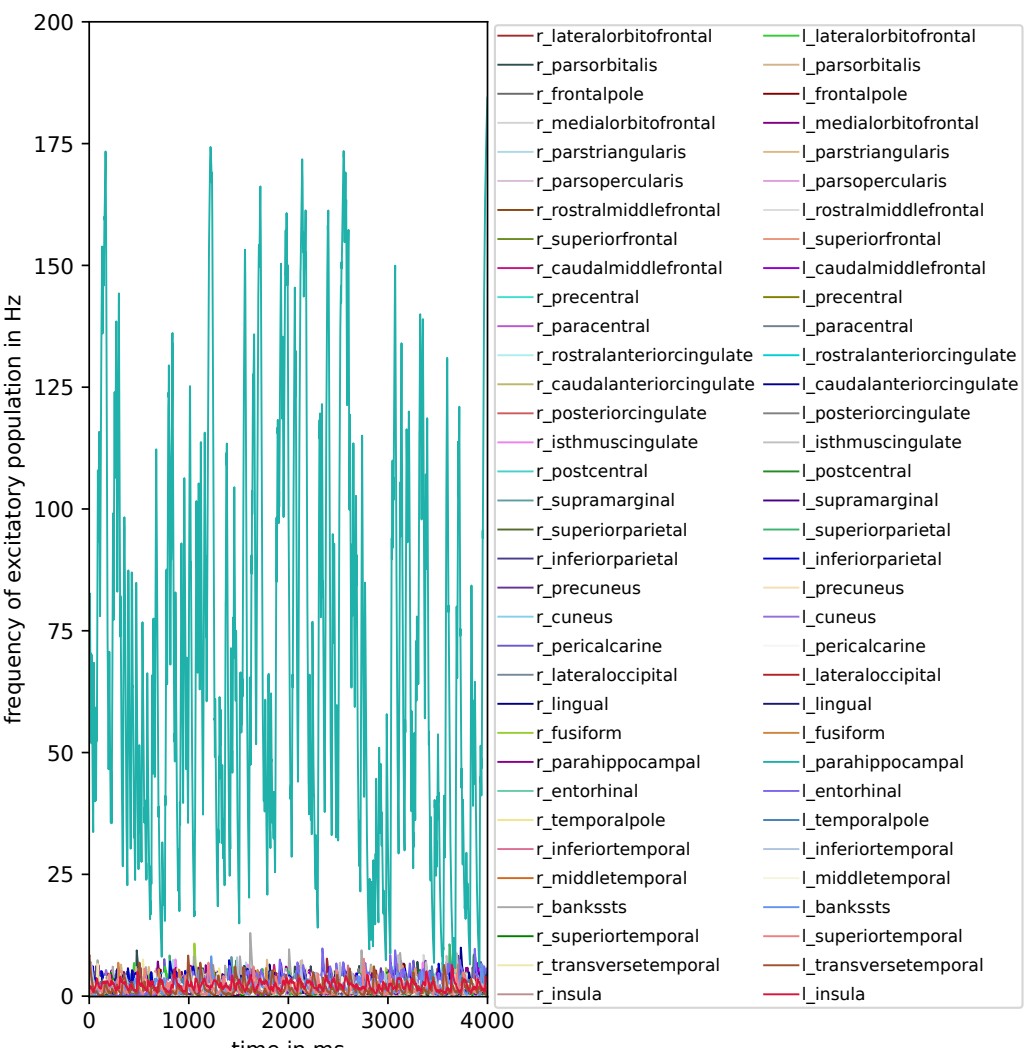

**Figure 7.** Time-series of the excitatory population firing rate for each brain region when action potential velocity is equal to 1 m/s and $g$: 0.4, $b_e$: 86, $wNoise$: $1 \times 10^{-4}$, $\tau_w$: 417, $a_e$: 0.0, $c_{F_{e,e}}$: $0.5 \times 10^{-3}$, $c_{F_{e,i}}$: 0.0, $c_{F_{i,e}}$: $0.5 \times 10^{-3}$, $c_{F_{i,i}}$: 0.0.

Apparently, the impact of speed is not significant unless it falls below or exceeds the value of 1.0 m/s, as displayed in Figure 8, where the coupling and spike-frequency adaptation are each modulated with a different value for speed. These results indicate that only for a speed of 1.0 m/s, is the maximum and variance in firing rates for both excitatory and inhibitory populations is maximal. When the speed increases, the firing rate does not exceed 20 Hz, which is similar to healthy brain dynamics. The first two graphs, which show

the mean firing rate for both populations of the column labeled "speed :1.0", also suggest that not all regions demonstrate an inclination to reach such elevated firing rates.

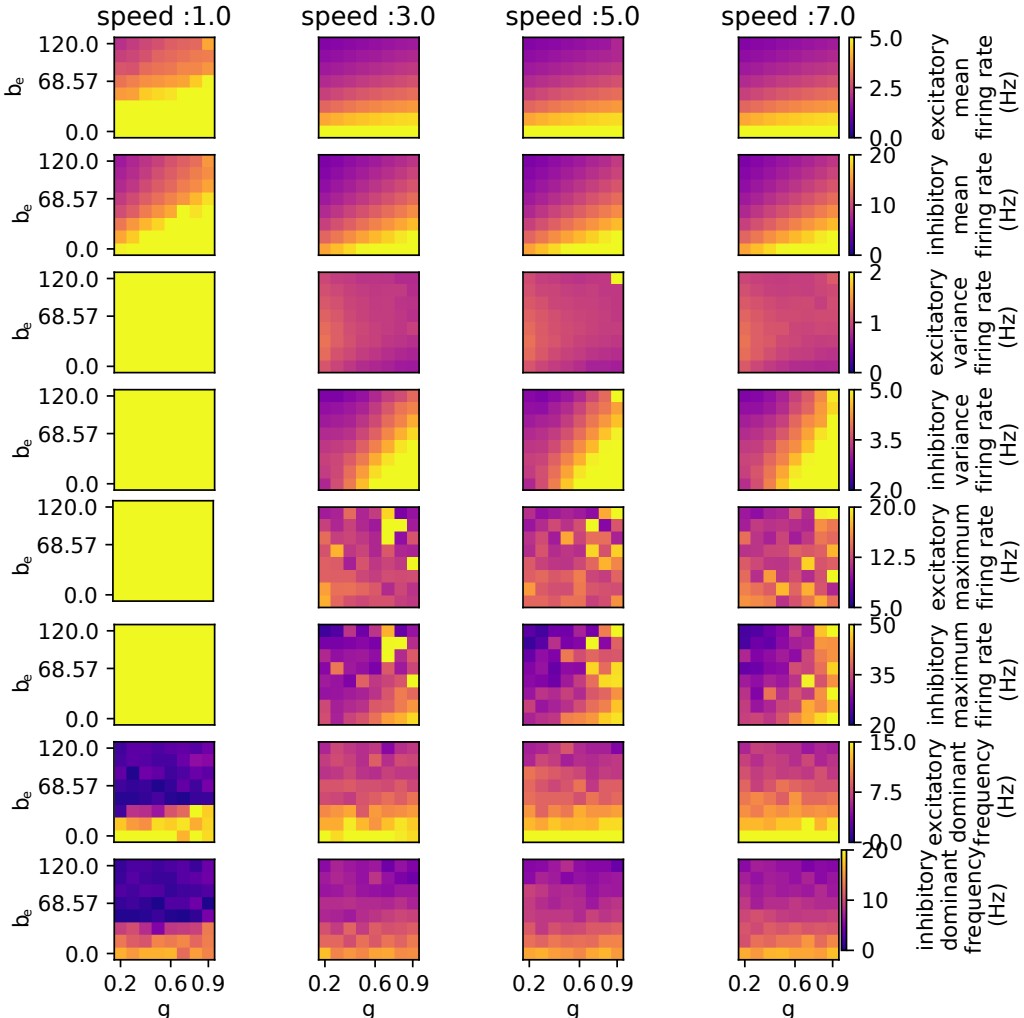

**Figure 8.** Parameter exploration for coupling and spike-frequency adaptation of excitatory neurons when action potential propagation speed is modulated. The graphs depicted in the [speed: 1.0] column indicate divergent behavior characterized by significantly higher firing rates compared to the other columns. The other parameters have the following values: *wNoise*: $1.0 \times 10^{-4}$, $\tau_w$: 417, $a_e$: 0.0, $c_{F_{e,e}}$: $0.3 \times 10^{-3}$, $c_{F_{e,i}}$: 0.0, $c_{F_{i,e}}$: $0.3 \times 10^{-3}$, $c_{F_{i,i}}$: 0.0.

### 3.3. Performance

In a previous study, the scaling relation between a CPU and GPU implementations was already performed [22]. The reported CPU implementation also makes use of the TVB-numba backend (an open-source Just-In-Time compiler for Python), but implements the Montbrio–Paxin–Rosin model [33], of which the scaling results are comparable to the TVB-AdEx CPU implementation. In this section, we report the execution time for the GPU simulation and analysis for the AdEx model only.

Strong and weak scaling are concepts used to evaluate the efficiency and performance of parallel computing systems. Strong scaling measures how well the application performs as the number of processors is increased while keeping the problem size fixed. Weak scaling assesses the performance as both the problem size and the number of processors are increased proportionally. The execution time analysis, plotted in Figure 9, indicates linear results for strong and weak scaling in relation to doubling the number of GPUs. The first graph displays the scaling behaviour for the simulation part, for which the overall

execution time remains under 600 s. There is a slight decrease in execution time when strong scaling is applied to the simulation part of the framework. This is due to the fact that the GPU is already optimal when fully occupied, making use of many threads to hide memory latency. Adding more GPUs and dividing the load makes the execution relatively less efficient because of a lesser thread count per GPU to hide latency.

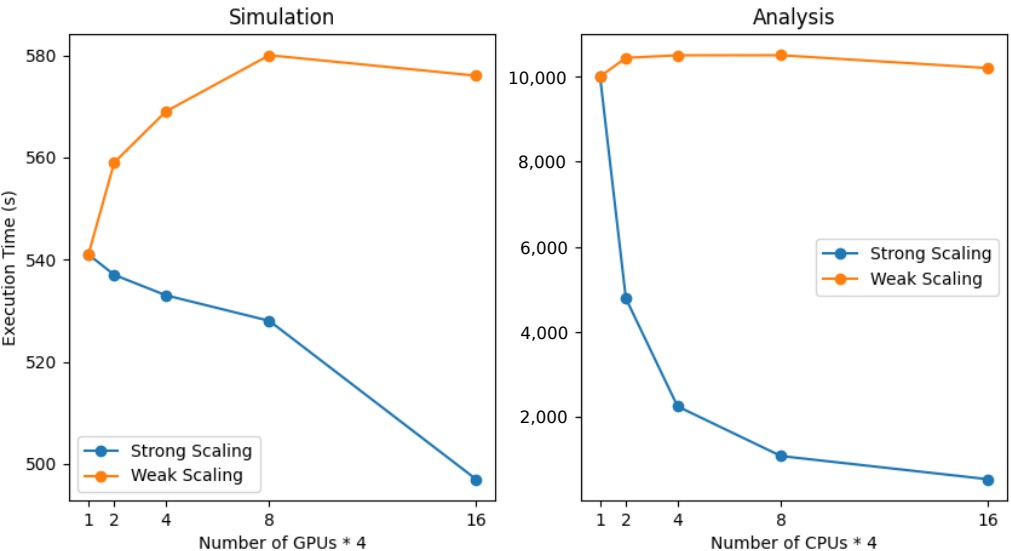

**Figure 9.** Scaling resources of strong and weak scaling results for the simulation and analysis parts of the framework, doubling the number of resources. The x-axis displays the number of nodes, each equipped with four GPUs and CPUs. The results are obtained on the Juwels Booster cluster located at the Forschungszentrum Jülich.

The second graph shows the execution time for the analysis part of the framework, which is conducted on the CPUs of the nodes. As expected, the analysis part benefits greatly from adding more resources, as run-time diminishes by half every time resources are doubled. The weak scaling indicates that the execution time for the analysis, which is the bottleneck of the framework, remains at a constant of approximately 10,000 s.

The temporal buffer maintains a record of the model states throughout the progression of a TVB simulation, incorporating signal propagation delays. The size of the temporal buffer is determined by the product of the number of states, the maximum length of the connectome, and the number of brain regions. Consequently, the temporal buffer constitutes the primary contributor to the GPU's memory footprint, rendering the GPU's scalability and maximum number of parameters contingent upon either the number of model states or the size of the connectome. Table 4 reports the memory utilization and number of concurrent unique TVB simulations for a single GPU in relation to the number of nodes as depicted in the scaling plots. The memory utilization for the weak scaling remains constant at 36,330 GB. The minimal simulation quantity consists of two instances of TVB, necessitating a memory allocation of 360 MB on the GPU. A CUDA device possessing a compute capability of 6.1 or higher, along with a memory capacity exceeding 360 MB, will have the capability to execute a concurrent set of TVB simulations.

The maximum number of resources of the cluster utilized for simulations is 384 nodes (out of 936 nodes), whereby each has four GPUs, totalling 1536 GPUs. This amount of resources enabled a parameter exploration of 25,165,824 concurrent TVB instances, each running simulations with different sets of parameters. The execution time for handling such a vast number of instances, encompassing both the simulation and analysis phases, remains well below the thresholds of 530 s and 10,000 s, respectively.

## 4. Discussion

In this work, we report the GPU implementation of the TVB-AdEx, adaptive exponential integrate-and-fire (AdEx) mean-field models adapted to The Virtual Brain (TVB) environment, previously used in studies on consciousness, linking mechanisms operating at microscopic scales to global brain dynamics. We explore the potential enhancements achievable through the utilization of MPI for vast GPU cluster parallelization and show the possibilities of increasing computational resources for complex brain models. The data generated from numerous parameter combinations can become overwhelming. We demonstrate that by conducting the analysis directly and organizing the outcomes in a database format, this framework enables oversight and becomes highly reusable. Furthermore, its modular architecture allows users to incorporate additional mean-field models for distributed processing across a compute cluster, especially for resource-intensive analyses.

Results on previously implemented models in the GPU framework, such as the Kuramoto [34], Wong-Wang [35], the Epileptor [36], and Montbrio–Paxin–9–Rosin [33], showing performant behaviour, have been previously reported [22]. With this work, we go one step further by providing a strategy to address more complex models, like the AdEx mean-field model, which can largely benefit from optimization of GPUs. The capability to compare the simulated output with (dynamical) functional connectivity renders this framework highly suitable, for instance, in the creation of virtual brain twins. These methods endow users with the capacity to discern connections between externally acquired data, whether empirical or simulated neuroimaging data like EEG, or even an externally obtained BOLD signal through the utilization of the GPU BOLD kernel.

As documented in [10], the multi-node implementation employing MPI, for the purpose of exploring a constrained parameter space in this model, allocated all unique parameter combinations or work-items to a single core out of the 128 cores available on the JUSUF supercomputer in Jülich, taking roughly 6 min of run-time for 5 s of simulated brain time. The GPU TVB-AdEx model takes about 8 min to compute 5 s of biological time for 16,384 work-items concurrently, increasing the resolution of the parameter space by 128 times for the simulation part of the framework. Upon scrutinizing the scalability of the implemented analysis tools, it becomes evident that this will also potentially benefit from a GPU implementation. This is exemplified by the observation that doubling the computational resources results in a reduction in run-time by half. Such an implementation would be well suited for the embarrassingly parallel characteristics of the analysis pipelines, accelerating the reconnaissance of the parameter space of this and other models even further. Plus, the same GPU pipelines could be equally useful for the rapid analysis of electroencephalography (EEG) and other time-series data. This is a priority for future research.

The Juwels Booster cluster at the Jülich Forschungszentrum has a total of 936 nodes. This substantial resource pool allows for the theoretical exploration of parameter spaces encompassing a staggering 61,341,696 possible configurations, all within a time-frame comparable to our current execution times. These impressive computational resources usher in a new era, one in which every single parameter of the model can be comprehensively examined, leaving no aspect unexplored. However, it is worth noting that new challenges and limitations emerge in this expanded landscape. For instance, the amount of memory required to store the vast amounts of information for plotting and analysis becomes a crucial consideration, i.e., the table which stores 65,536 rows (unique simulations) $\times$ 607 columns (analysis entries)—the results for a single node—requires 301 megabytes. If the time-series resulting from the 65,536 TVB simulations over 50,000 steps are to be preserved, the aggregate storage demand would escalate to 139.28 gigabytes. For a singular computational node, the corresponding memory requirement is estimated at 139.58 gigabytes. Extrapolating these to encompass all 984 nodes in use, the cumulative memory demand reaches 134 terabytes.

The sheer volume of data generated will necessitate more sophisticated methods of exploration and analysis, surpassing the capacity of individuals needed to manage it effec-

tively. In this regard, the utilization of specialized tools becomes pressing. One such tool that holds promise in navigating these immense data-sets is "Learning to Learn" (L2L) [37]. This automated machine learning framework is purpose-built for high-performance computing environments and is adept at employing gradient or evolutionary strategies to traverse expansive data generated by our framework. Furthermore, the HPC framework for TVB can serve as an initial step in the process of mapping the intricate properties of the parameter space. These preliminary insights can then be leveraged by Learning to Learn to intelligently identify and elucidate intriguing patterns within this expansive and complex space.

In this work, we reported our HPC implementations of multi-scale brain model simulation and analysis toolkits, taking a step in the direction of creating a complete workflow specifically designed to address metrics related to transitions in brain dynamics associated with varying degrees of consciousness. Thus, this comprehensive simulation and analysis framework distinguishes itself as a unique and valuable tool for advancing the development of virtual brain twins, not only within the realm of theoretical research, but also in the context of personalized clinical investigations for epilepsy, sleep, anesthesia, and disorders of consciousness.

**Author Contributions:** Conceptualization, M.v.d.V., S.D.-P. and J.S.G.; Data curation, M.v.d.V. and S.D.-P.; Formal analysis, M.v.d.V., L.K. and J.S.G.; Funding acquisition, A.D., V.J. and S.D.-P.; Investigation, M.v.d.V., L.K. and J.S.G.; Methodology, M.v.d.V., L.K., S.D.-P. and J.S.G.; Project administration, M.v.d.V., A.D., V.J. and S.D.-P.; Resources, A.D., V.J. and S.D.-P.; Software, M.v.d.V. and L.K.; Supervision, A.D., V.J., S.D.-P. and J.S.G.; Validation, M.v.d.V. and L.K.; Visualization, M.v.d.V. and L.K.; Writing—original draft, M.v.d.V., L.K. and J.S.G.; Writing—review and editing, M.v.d.V., L.K., A.D. and J.S.G. All authors have read and agreed to the published version of the manuscript.

**Funding:** The research leading to these results has received funding from the European Union's Horizon 2020 Framework Program for Research and Innovation under the Grant Agreements No. 945539 (Human Brain Project SGA3) and 101058516 (eBRAIN-Health). This research was also supported by the Helmholtz Joint Lab "Supercomputing and Modeling for the Human Brain". Open Access publication was funded by the Deutsche Forschungsgemeinschaft (DFG, German Research Foundation)—491111487.

**Institutional Review Board Statement:** Not applicable.

**Informed Consent Statement:** Not applicable.

**Data Availability Statement:** The original contributions presented in the study are included in the article, further inquiries can be directed to the corresponding author.

**Acknowledgments:** The authors thank David Aquilue, Nuria Tort-Colet, and Trang-Anh E. Nghiem for contributing analysis scripts. We also thank David Aquilue for his helpful discussion of the manuscript and for contributing data from his previous work [10] for a comparison with the GPU AdEx model reported here. We acknowledge the use of Fenix Infrastructure Resources, which are partially funded from the European Union's Horizon 2020 research and innovation programme through the ICEI project under the grant agreement No. 800858. The authors gratefully acknowledge the Gauss Centre for Supercomputing e.V. (www.gauss/centre.eu, accessed on 1 November 2023) for funding this project by providing computing time on the GCS Supercomputer JUWELS at Jülich Supercomputing Centre (JSC).

**Conflicts of Interest:** The authors declare no conflicts of interest.

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
