# Peer review of "Vast Parameter Space Exploration of the Virtual Brain: A Modular Framework for Accelerating the Multi-Scale Simulation of Human Brain Dynamics"

_applsci, doi:10.3390/app14052211_

Round 1

Reviewer 1 Report

Comments and Suggestions for Authors

The paper presents a GPU-based implementation of the TVB neuroinformatics platform, the TVB-HPC framework, a modular set of methods to implement the TVB-AdEx model. This framework allows extensive parameter space explorations and analysis of emergent dynamics, notably accelerating these simulations and, consequently, reducing computational resource requirements. The paper discusses the resulting comparison with other implementations of the same model and reports new parameter-space explorations never studied before.

First of all, I would like to commend the authors for their interesting and timely work. High-performance platforms are in great need in the neuroinformatics community, and TVB, despite being the most popular platform, has a serious need for accelerating techniques. This paper presents a solid contribution towards filling this gap.

However, the paper mentions but does not discuss, a problem that I consider of utmost importance: the similarity, or thereby lack of, concerning the reference CPU implementation. This is discussed in the two paragraphs on page 8, just below Equation 6, and I consider this point to be of extreme importance and discuss it in detail. According to the results in the manuscript, the best concordance achieved between the two implementations (CPU and the one presented here) is at best around 70~77% of the reference one. This is very worrisome, as the whole reliability of this new proposal is at stake. Why exactly is there such discrepancy, even when the two implementations of the SAME model should be equivalent, and the input parameters are? What are the ultimate reasons why I should trust a platform that does not give the same results as the original one? 75% accuracy may be enough for some purposes, but if the effect I am trying to discern is somewhat subtle, this 25% lack of accuracy may ruin my experiment completely! As you would understand, a future user needs to be certain that the platform, referring to the CPU vs GPU implementations, is robust and reliable enough. Also, the document reports that different datasets give results that are closer to the target one than the original parameters! Why do two different parameter sets give similar, and even better results than the ones used for the reference simulation? Is there some kind of "degeneracy" of the AdEx model that produces this lack of sensitivity?  Please, clarify.

Another point that is still unclear to me is the memory usage. The paper, at the very end, mentions this shallowly, but I would like to know what is the memory consumption for each of the simulations done. Line 538 mentions that storing information for analysis and plotting could be a "crucial consideration", but no figures are provided. Please, describe in detail. Also, add details about memory requirements, especially in the case of the use of delays, as the first paragraph on page 18 clearly states that this is a point to consider.

Some minor comments:
* The paper states that the authors with "double cross" contributed equally, but this symbol is not used anywhere!
* On page 3, line 98, the paper says that this WOULD ALLOW to run 40 million TVB instances, but later on it provides details of this case. Please, rephrase. 
* Line 138 reads "One of the main originality of this formalism is..." Please, check the phrasing of this sentence.
* Lines 210 and 211 on page 6 show a piece of code for computing the average of two observables, but not all observables are averaged. For instance, distributions could be accumulated and compared with the Kolmogorov-Smirnov statistic...
* Line 325 on page 8, mentions a case that is not presented in Table 3: g = 0.4, b_e = 80, E_{l,e} = E_{l,i} = −70. I guess b_e=72, right?
* I couldn't find a reference to Figure 5 in Section 3.2.3. Please, add.
* Section 3.3 discusses performance comparing strong and weak scaling, but these terms have never been defined before. Please, add a definition and make the paper self-contained.

Summing it all up, I think this is a VERY VALUABLE and INTERESTING paper, but there are some missing descriptions that I believe are crucial for future readers and, most importantly, future users of the framework.

Author Response

Thank you very much for taking the time to review this manuscript.Please see the attachment with a point-by-point response to the reviewer’s comments.

Reviewer 2 Report

Comments and Suggestions for Authors

The authors present an introduction to their GPU computing system for the TVB-AdEx model on large region-based mean fields. The presentation effectively captures essential features of the model and key results, as well as provides evaluations of the GPU computation over the traditional CPU-based computation. I think this is suitable for publication with only minor changes in the text. However, the authors use highly specialized terminology without much explanation. For example, some acronyms are not spelt out, such as AdEx. 

Minor comments:

1. Equation (3) and line 121: suggest to replace b with be for consistency.

2. Line 276: suggest to use other alphabets than a and b to avoid confusion.

3. Line 358: replace b_e with be.

Author Response

We express our gratitude to the reviewer for dedicating their time and effort to the review process. We have addressed the minor comments, spelt out acronyms and clarified highly specialized terminology.

Having identified a minor bug in the transfer function of the AdEx model, we have re-executed the simulations and subsequently revised the figures. 
Upon reevaluation, it is noted that the disparities are not substantial.
The most significant distinction is evident in Chapter 3.2.4, where the synchrony conditions are not as firmly identified as in the previous analysis.

Reviewer 3 Report

Comments and Suggestions for Authors

In this manuscript, the authors reported their TVB-HPC framework to implement the TVB-AdEx mean-field model on GPU. This framework preserves the stability and robustness of the TVB-AdEx model. They demonstrated the similarity of parameter combinations giving rise to patterns of functional connectivity between brain regions. They also reported multiple phenomena from their explorations of parameter space. This manuscript provided a comprehensive simulation and analysis framework to study human brain dynamics. 

This work is sufficient to be published in its present form. I don’t have additional comments or suggestions.

Author Response

We express our gratitude to the reviewer for dedicating their time and effort to the review process.

Having identified a minor bug in the transfer function of the AdEx model, we have re-executed the simulations and subsequently revised the figures. 
Upon reevaluation, it is noted that the disparities are not substantial.
The most significant distinction is evident in Chapter 3.2.4, where the synchrony conditions are not as firmly identified as in the previous analysis.

Round 2

Reviewer 1 Report

Comments and Suggestions for Authors

I think the authors answered all the questions and doubts I posed in my previous version quite satisfactorily. For me, this paper is ready for publication.